# Prestimulus functional connectivity reflects attention orientation in a prospective memory task: A magnetoencephalographic (MEG) study

**Stefano Vicentin**[1,2]*, **Giorgia Cona**[1,2], **Marco Marino**[1,3], **Patrizia Bisiacchi**[1,2], **Dante Mantini**[3], **Giorgio Arcara**[1,4]

**1** Department of General Psychology, University of Padua, Padua, Italy, **2** Padova Neuroscience Center, Padua, Italy, **3** Movement Control and Neuroplasticity Research Group, KU Leuven, Leuven, Belgium, **4** IRCCS San Camillo Hospital, Venice, Italy

* stefano.vicentin@unipd.it

## Abstract

Prospective Memory (PM) is the ability to encode an intention in memory and retrieve it at the right time in the future. After the intention is formed, it must be maintained in memory while simultaneously monitoring the environment until the occurrence of the stimulus associated with its retrieval. Therefore, monitoring and maintenance processes must work in conjunction to subserve PM processing (monitoring/maintenance phase). Several brain regions play a role in PM, such as the anterior prefrontal cortex, inferior parietal lobules, and precuneus. Notably, these regions belong to different brain networks and are differently involved depending on the memory and attentional requests of the PM task. In this study, we investigate the neural bases of PM from a network perspective, using functional connectivity (FC) analysis to identify the networks involved in the attentional and memory mechanisms underlying PM. To this end, we analyzed MEG data collected in two different PM conditions, enhancing either the monitoring (i.e., attention) or the maintenance (i.e., memory) loads of the PM task. To disentangle the neural correlates of these mechanisms from other processes occurring after stimulus presentation, the analysis focused on the prestimulus time window (monitoring/maintenance phase). The monitoring-load condition was characterized by increased inter-network FC of the Dorsal Attention Network (DAN) in the alpha band, a marker of increased top-down monitoring. In contrast, the maintenance-load condition was associated with increased connectivity of the Ventral Attention Network (VAN) with the FrontoParietal Control and the Default-Mode Networks (FPCN and DMN, respectively). Additionally, response times were found to correlate with prestimulus alpha connectivity of different networks in the two conditions. These differences in connectivity within and between networks support the hypothesis that different networks (DAN, or VAN and DMN) and mechanisms (top-down or bottom-up, respectively) are involved in PM processing depending on the features of the PM task.

**Data availability statement:** Functional connectivity matrices for each participant and condition (together with the experimental paradigm applied in this study) are available in the OSF repository (https://osf.io/5whb7/).

**Funding:** The author(s) received no specific funding for this work.

**Competing interests:** The authors have declared that no competing interests exist.

## Introduction

Prospective Memory (PM) refers to the ability to remember to perform an intention at the occurrence of a specific event or moment in the future [1]. PM processes are both frequent and essential in everyday activities [2]: common examples include remembering to take prescribed medication at the right time, updating a colleague after a meeting, or charging the phone upon finding a charger. Prospective remembering comprises several stages [3]: First, the intention must be formed and stored in memory along with the information associated with its retrieval (encoding phase). In the previous example, we might notice that our phone battery is low and decide to charge it as soon as possible. The stimulus or event associated with the intention (in this case, finding the charger) is defined as the *PM cue*. Subsequently, the intention must be retained in memory until the occurrence of the PM cue (maintenance phase). During this phase, individuals often engage in unrelated activities, known as *ongoing activities* (e.g., finishing our work activities) [4]. Simultaneously, attentional resources must be directed both to the environment (external attention), monitoring for the occurrence of the PM cue, and towards internal contents (internal attention) to uphold the intention in memory (e.g., "I need to charge the phone—where can the charger be?") [5]. The balance between internal and external attention is essential for successfully retrieving the intention once the PM cue is detected (retrieval phase), comparing external stimuli with the predetermined idea of the PM cue. Finally, once the intention is correctly recalled from memory, it can effectively be carried out (execution phase).

Several models have been proposed to elucidate the dynamic interplay between attentional and memory mechanisms underlying intention maintenance and prospective remembering. The Preparatory Attention and Memory theory (PAM) [6] emphasizes the anticipatory nature of attentional and memory processes, focusing on their roles during the maintenance phase. In this phase, cognitive resources are allocated to monitor the environment for cues relevant to intention retrieval (preparatory attentional mechanisms). Concurrently, preparatory memory processes maintain the intention and the associated information active in memory, facilitating its prompt retrieval and execution upon detection of the PM cue. In line with this framework, and to enhance clarity, we will hereinafter refer to the maintenance phase as the *monitoring/maintenance phase*. The Multiprocess Framework [7,8] hypothesizes the existence of two mechanisms that can be employed to retrieve intentions from memory depending on the monitoring and maintenance demands of the PM task. The first mechanism, *strategic monitoring*, involves a series of top-down processes deputed to actively maintain the intention in memory and to continuously monitor the environment searching for the PM cue. This mechanism is preferentially activated when the PM and ongoing tasks requests enhance the memory and attentional load (e.g., multiple intentions to maintain, covert PM cues). The second mechanism, *spontaneous retrieval*, occurs automatically when monitoring and maintenance demands are lower, allowing the intention to be effortlessly triggered by the PM cue itself.

The presence of distinct mechanisms underlying prospective remembering has been confirmed by several neuroimaging studies [9–12]. In general, PM processing was found to rely on the activation of multiple brain regions, including the anterior prefrontal cortex (aPFC), the frontoparietal network (FPN), the cingulate cortex, the insular and medial temporal regions, and subcortical nuclei such as the thalamus and putamen. However, opposite patterns of activation have been observed in the medial and lateral aPFC and between the ventral and dorsal portions of the FPN in relation to PM task features (such as salience and focality) or during specific PM phases [13–16]. Briefly, focality refers to the extent to which the processing demands of the ongoing task overlap with those of the PM tasks (e.g., asking to detect

a specific number while doing an arithmetical task vs. asking to detect all numbers starting with the letter "T"), whereas salience refers to the perceptual difference between the ongoing stimuli and the PM cue (e.g., a picture of a yellow taxi placed in a snowy landscape vs. one set against the busy traffic of New York City). Focal and salient stimuli have been consistently found to induce the activation of bottom-up processes (spontaneous retrieval), mostly relying on the ventral FPN, whereas nonfocal and non-salient stimuli induced the activation of the dorsal portion of the FPN [14,17]. Concerning PM phases, the dorsal FPN was preferentially engaged during the monitoring/maintenance phase, whereas the ventral FPN during retrieval. Notably, the dorsal FPN corresponds to the Dorsal Attention Network (DAN), which is associated with top-down processing and external attention [18,19]. In contrast, the ventral FPN shares its core nodes with the Ventral Attention Network (VAN), which is involved in bottom-up processes triggered by stimulus presentation [20,21]. Interpreting these findings in the light of the Multiprocess Framework, Cona and colleagues proposed the Attention to Delayed Intention model (AToDI model) [12]. In this view, the selective activation of the DAN reflects the preferential activation of top-down mechanisms (strategic monitoring) to perform the PM task, whereas the involvement of the VAN reveals the involvement of more bottom-up processes (spontaneous retrieval) [22,23].

The studies investigating the neural bases of PM mostly employed neuroimaging techniques such as functional magnetic resonance imaging (fMRI) and positron emission tomography (PET) [9,24], techniques which excellent spatial resolution enabled to investigate the functional roles of specific subregions (such as the medial and lateral aPFC) and subcortical nuclei (thalamus, putamen). However, these techniques have limited temporal resolution, preventing the examination of the rapid neural oscillations associated with PM processing. In contrast, electroencephalography (EEG) and magnetoencephalography (MEG) offer a resolution in the order of milliseconds, allowing to observe the rapid variations in the transient activity associated with cognitive processes such as PM. These techniques also enable the separation of neural oscillations into different bands of interest based on their frequency, as neural activity occurring at different frequencies has different functional roles. For instance, the few studies on PM to collect neurophysiological (MEG or EEG) data in the time-frequency domain [25–27] focused on the theta (5–7 Hz) and alpha (8–12 Hz) frequency bands, due to the role of theta activity in memory processes such as working memory and internal processing [28,29], and the association between alpha oscillations with attentional mechanisms and monitoring processes [30,31].

MEG, in particular, offers excellent temporal resolution paired with good spatial resolution, which can be further enhanced by combining it with the individual structural MRI. Cona and colleagues [23] employed this technique to investigate the neural correlates of the monitoring and maintenance requests of PM. The authors designed a paradigm composed of three different conditions: an ongoing task presented alone (*Baseline* condition), and two conditions in which the same task was proposed together with some prospective instructions. The two PM tasks were designed to enhance either external monitoring attention (*Monitoring-load* condition) or memory demands (*Maintenance-load* condition). Comparing neural oscillations in the three conditions, the Monitoring-load condition presented a pattern of decreased alpha activity across regions of the DAN (frontal eye fields, dorsal parietal regions), whereas the Maintenance-load block showed increased theta activity over regions of the VAN. Notably, in this latter condition theta activation was also observed in regions of the Default Mode Network (DMN), a network associated with working memory and internal contents processing [32,33]. The authors interpreted these results within the ATODI model framework [12] suggesting that modulating the maintenance and monitoring demands of the PM task induced the involvement of different mechanisms and networks. Namely, increased monitoring demands triggered

attentional mechanisms, which were reflected in the increased involvement of the DAN. In contrast, higher levels of maintenance demands (coupled with lower monitoring demands) triggered internal attention and memory processes relying on the activation of VAN and DMN regions.

Despite accumulating evidence that different networks may be engaged in PM processing depending on the features of the PM task, prospective remembering has never been investigated at the network level. As a result, direct evidence linking specific networks to mechanisms such as strategic monitoring and spontaneous retrieval remains lacking. This gap can be addressed using a Functional Connectivity (FC) approach. In neuroscience, FC is defined as the statistical dependencies between spatially distinct neurophysiological events [34], which enable the identification of co-activation patterns among distant brain regions. These co-activations suggest that the regions work together to execute specific processes [35]. Notably, FC allows studying neural oscillations also in the prestimulus time window [36], making it a powerful tool for studying cognitive processes like PM, in which maintenance and monitoring mechanisms are not tied to the stimulus presentation but instead remain active throughout the entire task (monitoring/maintenance phase). Furthermore, this PM phase is characterized by the presence of monitoring and maintenance processes in the absence of other confounding factors which inevitably occur after stimulus appearance, such as the elaboration of the ongoing task [6]. For these reasons, we believe that the investigation of PM through the analysis of the prestimulus time window offers a unique opportunity to deepen our understanding of the mechanisms underlying prospective remembering.

## The present study

This study employs a novel methodological approach proposed by Samogin et al. [37] to investigate functional connectivity (FC). Specifically, two measures—average within-network connectivity (IntraNC) and average between-network connectivity (InterNC)—were extracted for each network of interest, compared between networks, and analyzed across experimental conditions. This approach has been previously used to identify distinct connectivity modulations associated with specific cognitive processes, such as motor learning and face processing [38,39], as well as emotional dysregulation traits [40]. In this study, we applied this method to a dataset previously collected by Cona and colleagues [23], who examined post-stimulus neural oscillations in three prospective conditions: Baseline, Monitoring-load, and Maintenance-load. In contrast, this study focused on the prestimulus time window (the 1000 ms prior to stimulus presentation) to explore connectivity differences during the monitoring/maintenance phase associated with the memory and attentional demands of the PM tasks. FC measures were computed focusing on the networks known to play a role in PM processing: the DAN, VAN, and DMN [12,23]. A fourth network, the Frontoparietal Control Network (FPCN), was also included for its role in top-down control processes and mediation of communications between the DAN, DMN, and the VAN [29,41,42]. Thus, IntraNC and InterNC were measured for each of these networks and compared within and between each condition, as in [37].

Our first hypothesis predicts differing levels of FC between the Baseline and prospective memory (PM) conditions. In the Baseline condition, prestimulus activity is expected to reflect only the preparation for the ongoing task. In contrast, in the PM conditions, this interval also corresponds to the monitoring/maintenance phase. Therefore, we hypothesize higher levels of both IntraNC and InterNC in the alpha band in the PM conditions, reflecting the increased attentional demands directed toward both the intentions and the environment for detecting the PM cue, consistent with the ATODI model. The second hypothesis concerns distinct connectivity patterns characterizing the two PM conditions. In the Monitoring-load condition

we expect increased involvement of the DAN, consistent with its role in the monitoring/ maintenance phase of PM tasks. Specifically, heightened InterNC between the DAN and other networks in the alpha band is anticipated, reflecting the integration of attentional and cognitive control processes required for strategic monitoring. Conversely, DAN involvement is expected to not differ between the Baseline and the Maintenance-load conditions, as its PM demands induce reliance on spontaneous retrieval, a mechanism occurring after stimulus presentation [7]. Instead, the Maintenance-Load condition is predicted to show increased connectivity in the theta band within and between the DMN and VAN, reflecting higher memory demands and reliance on bottom-up processes. By examining intraNC and interNC during the monitoring/maintenance phase across different PM conditions, this study aims to shed light on the specific mechanisms underlying maintenance and monitoring processes underlying PM, providing new insights into the neural underpinnings of this complex and crucial mechanism, and offering innovative methods to unravel them.

## Materials and methods

### Data acquisition

Data were collected from 21 young right-handed healthy participants (age: 24.67 ± 2.24 years, range 21–29). The sample size was determined in the previous study employing this dataset [23] using G*Power to calculate the power analysis and was consistent with the numerosity of other studies investigating FC with the same methodologies [37,38]. The study was conducted following the guidelines of the Helsinki Declaration and received approval from the local ethics committee (Ethics Committee for clinical sperimentation of Venice, approval n. 1013/IRCCSsancamillo). Exclusion criteria included history of epileptic seizures, neurological or psychiatric disorders, metal implants in the body, acoustic implants, non-removable piercings or metal accessories. Participants took part voluntarily to the study and did not receive any form of compensation. All Participants provided informed consent before the beginning of the procedure. Data were acquired at the IRCCS San Camillo hospital between the 11th of October and the 17th of December 2018. The experimental procedure consisted of two separate sessions. First, neurophysiological data were acquired using a 275-channel MEG system (CTF-MEG). The MEG preparation session involved the placement of three head coils at anatomical landmarks (left and right preauricular points, nasion) for head localization and co-registration with anatomical images. Additionally, six amagnetic electrodes were used for auxiliary recordings: four were positioned around the eyes (two for detecting vertical eye movements and blinks, and two for horizontal eye movements), and two were placed on the left and right scapulas to capture cardiac activity. The positions of the head coils and the participant's head shape were digitized using a Polhemus Fastrak system to enable precise co-registration. After the preparation, participants were seated comfortably inside a Faraday cage and given instructions about the experimental tasks. They were informed that they would be monitored and could communicate at any time during the session. Participants were also assured they could request a break or terminate the experiment if they felt uncomfortable in the MEG room. During data acquisition, MEG signal was recorded continuously at a sampling rate of 1200 Hz, with a hardware anti-aliasing low-pass filter applied at 300 Hz.

In the second session, neuroimaging data were collected using a 1.5 Tesla Philips Achieva MRI scanner. Specifically, a T1-weighted whole-head anatomical image was acquired for each participant (repetition time [TR]: 8.3 ms, echo time [TE]: 4.1 ms, flip angle: 8°, acquired matrix resolution [MR] = 288 × 288, slice thickness [ST] = 0.87 mm). These images were used to create individual head models for MEG source reconstruction. Preprocessing of MRI data was performed using FreeSurfer [43] and included skull stripping, segmentation of subcortical

## Task procedure

The experimental paradigm was the same employed in two previous studies [22,23] to which the reader is referred for further details. Briefly, the experiment was administered through the Psychopy software (version 1.85.2) [44] and consisted of three different blocks, each composed of 120 trials (a visual representation of the three blocks is displayed in Fig 1). In the first block, defined as the Baseline condition, participants were asked to perform a Lexical Decision Task (LDT). LDT requires to judge as quickly and accurately as possible the lexicality of each presented string of letters (i.e., if they correspond to a word or not). The other two blocks consisted of two PM conditions in which two different PM tasks were added to the LDT, which in these conditions served as the ongoing task. The two PM tasks differed in terms of specific PM features that were modulated to enhance either the maintenance (thus, attention directed to the intention in memory) or the monitoring (external attention to detect the PM cue) load. Specifically, PM features like the number of intentions to maintain, salience, and focality were manipulated. Indeed, while increasing the number of intentions has been found to enhance maintenance load [22,45], focality and salience present an inverse relationship with external monitoring, as non-focal and less salient PM cues require higher levels of external attention [16,46]. In both PM tasks, a total of ten PM cues were presented among the 120 trials (~ 10% of the total stimuli). PM cues were placed in pseudo-random positions, the first occurring

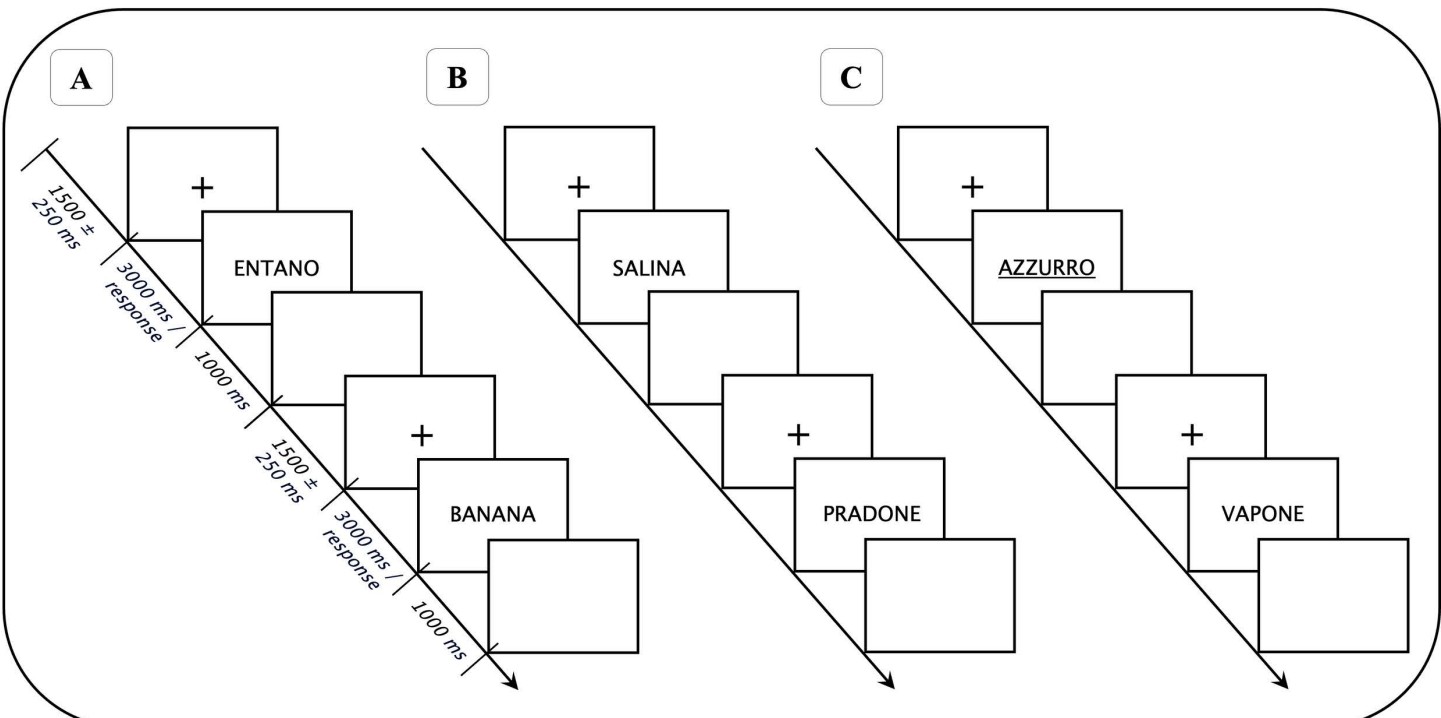

**Fig 1. General structure of the experimental paradigm.** Independently from the condition, the strings of letters for the LDT were presented until a response was given, or for a maximum of 3000 ms. Hence, a blank screen was presented for 1000 ms, followed by a fixation cross displayed for an interval of 1500 ± 250 ms. A) In the Baseline condition, participants only had to perform the LDT. B) In the Monitoring-Load condition, participants were instructed to perform the LDT and to press a different button when the syllable "PRA" appeared. C) In the Memory-Load condition, participants were asked to perform the LDT and to to press three different buttons at the presentation of three underlined words (AZZURRO, GRIGIO, and ROSSO).

after at least ten ongoing trials, and consecutive PM cues separated by at least eight ongoing trials. At the beginning of the experiment, a practice block involving only the ongoing task (five words/nonwords trials) was administered.

In the Maintenance-load PM task, participants were instructed to maintain three different intentions in memory (thus enhancing the attention to internal contents), each associated with a specific underlined word (focal and salient PM task, thus reducing monitoring demands and promoting spontaneous retrieval). Conversely, in the Monitoring-load PM task, a single prospective instruction was given (low maintenance load), asking participants to detect a specific syllable. In this condition, the PM task was non-focal (the PM and the ongoing task required to process stimuli at different levels) and non-salient, as the PM cue did not differ perceptually from the other trials. This type of PM task is known in the literature to promote the reliance on strategic monitoring [8,47] enhancing external attention and the top-down processing of the task.

To avoid commission errors (i.e., performing the PM task even when no longer requested) and according to the literature on PM, the Baseline condition was always presented first. The order of presentation of the two PM tasks, instead, was counterbalanced across participants.

## MEG preprocessing and source estimation

MEG preprocessing was performed using Brainstorm [48]. First, data were down-sampled to 600 Hz and biological artifacts associated with cardiac activity and ocular movements were removed from the recordings using the Signal-Space Projector (SSP) algorithm, based on the signal collected through the cardiac and ocular electrodes. The resulting signal was then segmented into epochs (from 1.5s before the ongoing stimulus presentation to 1.5s after), considering only the ongoing trials with a correct response (Baseline: 97%; Monitoring-Load: 96%; Maintenance-Load: 98%). Thus, epochs were visually inspected to exclude the ones presenting artifactual signals from the following analyses. For source estimation, the individual 3D T1 MRI was segmented using Freesurfer following the recon-all routine [49] and then aligned with each participant's neurophysiological data using the digitized position of the head coils and the head shape created with the Polhemus Fastrak system. The forward model was calculated using the overlapping spheres method [50], and the inverse problem was solved using the Minimum Norm Estimate. Noise covariance was calculated from 3 minutes recordings of the empty room noise, collected right after each participant's experimental session and preprocessed following the same pipeline.

## Functional connectivity analysis

Functional Connectivity (FC) was assessed using the Amplitude Envelope Correlation (AEC) method, which has been consistently demonstrated as a reliable approach to calculate connectivity from MEG data [51,52]. To investigate FC between brain regions considering both their anatomical and functional properties, we followed the procedure described in Amico et al. [53] and employed in [54,55]. Namely, the brain surface was divided into 148 parcels (74 per hemisphere) according to the Destrieux Atlas [56]. Thus, each of these parcels was assigned to the corresponding functional network according to the Yeo classification [57,58] based on the position of their centroid. The correspondence between each region and network is displayed in the S1 Table (Supporting Information). Among the seven Yeo networks, the analysis focused on the ones that have been shown to play a role in PM processing (the DAN, DMN, and VAN) and on the FPCN, due to its role as a mediator in the communication between the considered networks [59,60]. To assess the robustness of the results, the same analysis was also performed on all the seven Yeo networks, and the results are reported in S1 and S2 Figs (Supporting Information).

Connectivity values were calculated in the [−1000:0] time window of each trial and then averaged, separately for the theta (4–7 Hz) and alpha (8–13 Hz) frequency bands. AEC was computed pairwise between nodes, resulting in two 148x148 matrix of values (one for each frequency band). Signal from each parcel was orthogonalized to reduce leakage related to volume conduction, following the procedure available in Brainstorm. Signal from all the vertices of each parcel were summarized using a Principal Component Analysis (PCA). Pairwise comparisons between parcels were performed on this PCA-extracted signals. Consequently, the 148x148 AEC values were categorized according to the Yeo network associated with each region, and the values associated with the four networks of interest were extracted. This procedure (depicted in Fig 2) led to the creation of a 4x4 matrix for each participant, condition, and frequency band.

The clustering of the Destrieux regions according to the Yeo networks enabled the investigation of connectivity at the network level, allowing to measure FC within and between specific networks (each associated with specific functional meanings) and to compare these values between conditions. To this end, following the procedure proposed by Samogin and colleagues [37,61] we computed two values from each network of interest: the intra-network connectivity value (IntraNC), corresponding to the average of the connectivity values between the regions belonging to the same network, and the inter-network connectivity value

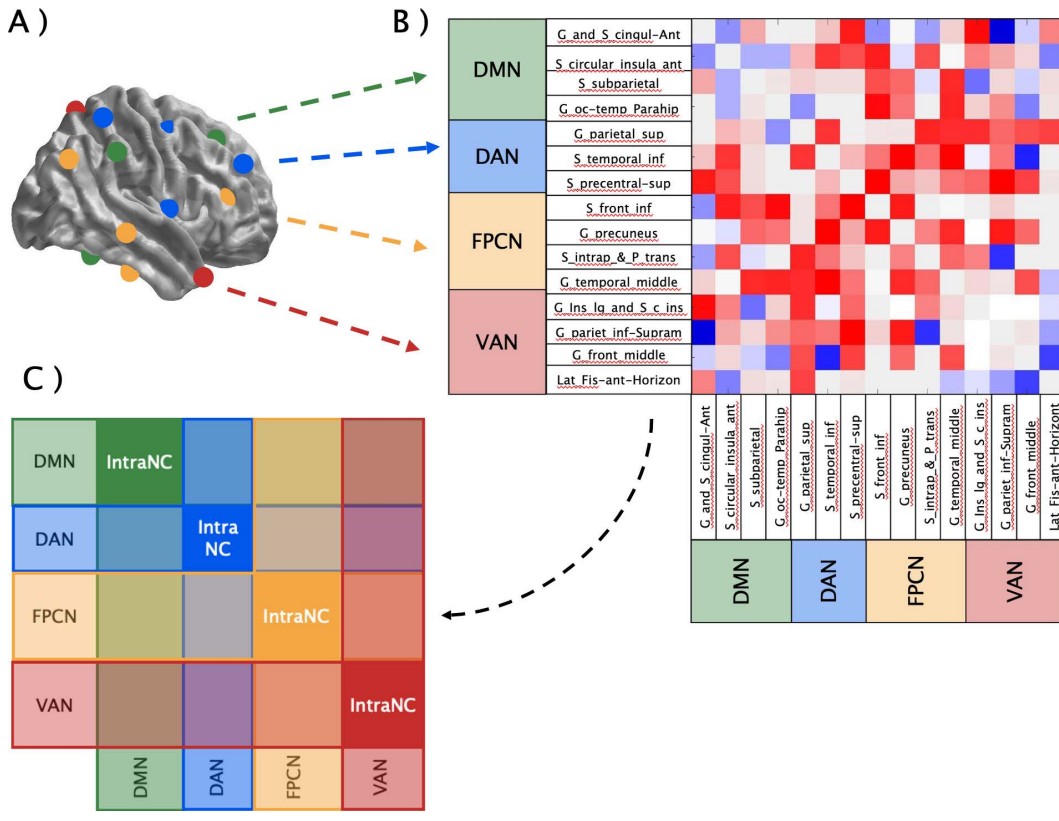

**Fig 2. Schematic representation of the pipeline followed to calculate IntraNC and InterNC values for each network.** A) The Destrieux regions belonging to the networks of interest (DMN, DAN, FPCN, and VAN) were selected to create a matrix of the AEC values between them; B) Brain regions were associated to the corresponding network, grouping the ones belonging to the DMN (green), DAN (blue), FPCN (yellow) and DAN (red); C) values of connectivity between regions belonging to the same network (IntraNC) and the ones between each pair of networks (InterNC) were averaged to obtain a 4x4 connectivity matrix at the network level.

(InterNC), corresponding to the average connectivity between all the regions of a specific network and all the regions belonging to the other considered networks [62,63]. To ensure proper normalization, we applied the fisher z-transformation (atanh) to the correlation values prior to averaging. Thus, IntraNC and InterNC values were compared between conditions using two different approaches [37,63]. First, for each network, we compared its IntraNC value to its InterNC value, separately for each condition. Then, for each network, we compared IntraNC and InterNC values between conditions, to identify networks showing significant changes in segregation (greater connectivity within the network) or integration (greater connectivity with other networks) depending on the task. Both approaches employed two-tailed paired t-tests with false discovery rate (FDR) correction for multiple comparisons, setting the significance level of the corrected p-value (thereinafter, $q$) to q < .05 [64].

## Correlation between behavioral and FC measures

Behavioral data were analyzed in a previous study, to which we refer the reader for further details [23]. In this study, we focused on analyzing the relationship between functional connectivity (FC) patterns and task performance to investigate whether distinct connectivity configurations in the prestimulus time window of different prospective memory (PM) tasks are associated with their behavioral outcomes. To achieve this, we paired the InterNC and IntraNC FC values of each network of interest with the corresponding Accuracy (ACC) and Response Times (RT) separately for each condition (Baseline, Monitoring-Load, and Maintenance-Load PM tasks) and frequency band (alpha, theta). Given that the Shapiro-Wilk test for multivariate normality indicated significant deviations from normality across all conditions (p < .001), we employed Spearman's Rho, a non-parametric correlation measure, to assess the relationships between prestimulus FC and behavioral performance.

## Comparison of power spectra

Increased connectivity between two regions (or networks), as measured by AEC, reflects concurrent modulation of their activity, but it does not distinguish whether this corresponds to co-activation or co-deactivation [65]. To address this issue, we conducted a Fast Fourier Transform (FFT) to estimate power spectra in each frequency band and network of interest, enabling direct comparison between conditions. Specifically, we calculated the raw power level for each node individually, applied a log-transformation to normalize the data, and then averaged it across the nodes within each network. This procedure was performed separately for each condition and frequency band to assess how power varied between conditions within each network. To analyze these differences, we conducted a 3 (conditions) × 4 (networks) repeated measures Analysis of Variance (ANOVA), with results corrected for multiple comparisons using the Bonferroni correction. This approach not only identified significant differences in power across conditions and networks but also provided insights into the interactions between these factors. Importantly, findings from this analysis complemented and confirmed the results obtained from the FC analysis.

## Results

### Intra- vs inter-network connectivity

In the alpha frequency band, the VAN showed significantly higher levels of InterNC compared to IntraNC in all conditions (Baseline, Monitoring-load, Maintenance-load; q < .005), indicating a tendency of the regions of this network to communicate more with regions of other networks than among themselves, regardless of the prospective load (Fig 3). The DAN, on the other hand, displayed this pattern of outward-oriented connectivity selectively in

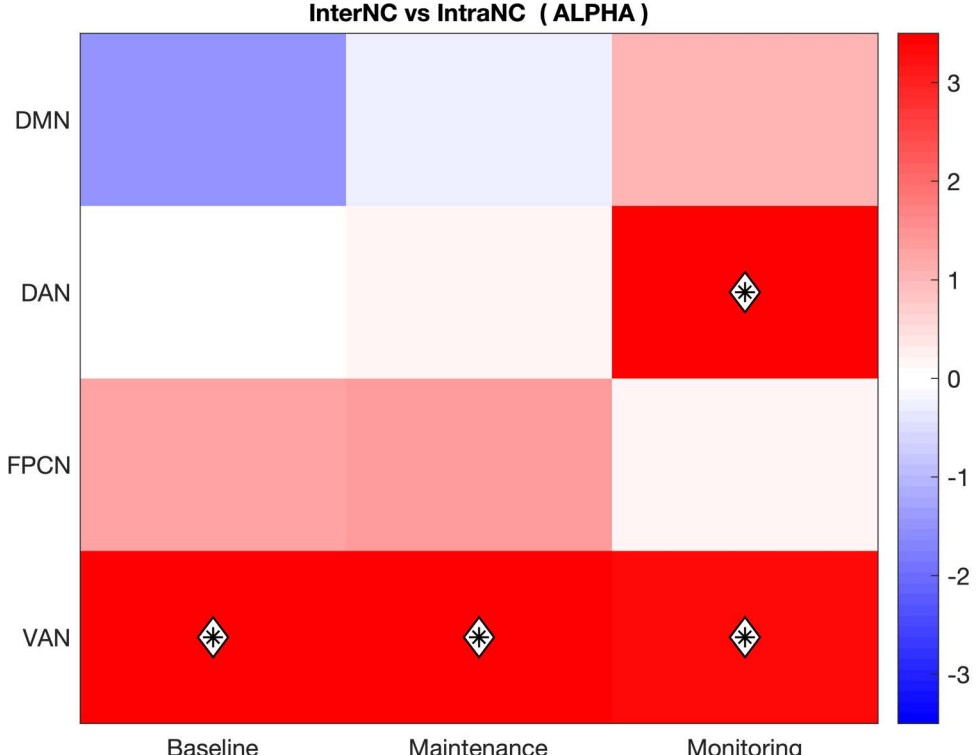

**Fig 3. Networks presenting significant differences between InterNC and IntraNC in the alpha band.** Differences significant for p < .05 are displayed as black asterisks, whereas those significant for q < .05 are marked with a white diamond. The scale on the left represents t-values derived from statistical comparisons. DMN: Default Mode Network; DAN: Dorsal Attention Network; FPCN: FrontoParietal Control Network; VAN: Ventral Attention Network.

the Monitoring-load condition (q < .001). While the higher InterNC of the VAN across all conditions may reflect bottom-up preparation to elaborate the upcoming stimulus, independently from its nature, the presence of the same effect over DAN regions selectively in the Monitoring-Load condition suggests a more top-down, strategical organization to detect less distinguishable PM cues.

In the theta frequency band, the only network showing significantly higher InterNC compared to IntraNC was the VAN in the Baseline condition, which displayed a trend of higher connectivity with the other networks than within its regions (p = .024). However, this result failed to reach significance once it was corrected for multiple comparisons (Fig 4).

## Comparisons across conditions

Comparing FC among the three conditions in the alpha band, significant differences between the PM conditions and the Baseline condition were found (Fig 5). Namely, the addition of a monitoring-enhancing PM task to the ongoing task was reflected in a widespread increase of connectivity, both between and within networks (q < .05). The only exception was the IntraNC in the DAN, in line with the enhanced InterNC of this network in the Monitoring-load condition reported in the previous paragraph (Fig 3) and with the hypothesis of a generalized "strategic preparatory state" for detecting less salient PM cues. Compared to the Baseline, in the Maintenance-load condition the VAN showed higher connectivity with the DMN (p = .007; q =.045) and the FPCN (p = .001; q =.045), both of which are associated with the processing of

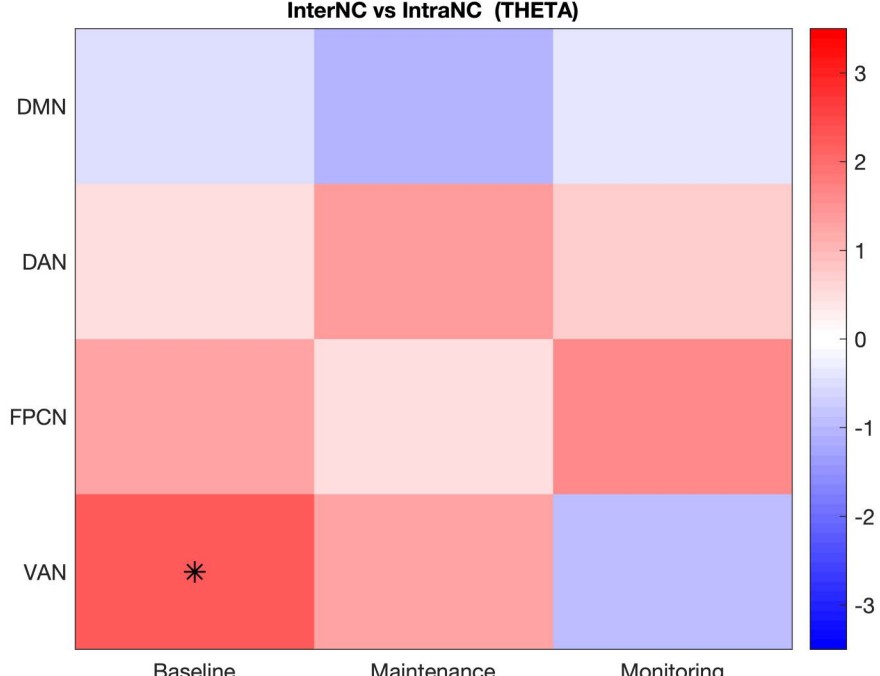

**Fig 4. Networks presenting significant differences between InterNC and IntraNC in the theta band.** Differences significant for p < .05 are displayed as black asterisks, whereas those significant for q < .05 are marked with a white diamond. The scale on the left represents t-values derived from statistical comparisons. DMN: Default Mode Network; DAN: Dorsal Attention Network; FPCN: FrontoParietal Control Network; VAN: Ventral Attention Network.

internal representations (Fig 5B). Hence, this connectivity pattern may indicate an increased focus on maintenance processes and preparation to select the correct intention. The DMN and VAN also showed a trend of increased IntraNC (DMN: p = .033; VAN: p = .011), but these results failed to reach significance once corrected for multiple comparisons. Finally, the comparison between the Maintenance- and the Monitoring-load conditions revealed differences mainly related to the DMN (Fig 5C). Specifically, in the Monitoring condition this network resulted to be significantly more connected with all the other considered sets of regions (q <.05). Furthermore, higher levels of IntraNC were found in this condition within the regions of the DMN (p = .023; q = .046). and the FPCN (p = .014; q = .046). This pattern of increased connectivity in the Monitoring-Load condition likely reflects heightened attentional resources directed toward the intention and allocated to upholding and refreshing the representation of a challenging-to-detect PM cue during the prestimulus time window, consistent with the engagement in strategic monitoring processes to prepare for intention retrieval. In contrast, in the Maintenance-Load condition, intention retrieval processes allegedly occur after stimulus presentation to recall the correct intention among multiple associated cues.

In the theta frequency band, the comparison between conditions highlighted some tendencies of increased connectivity in the two PM tasks (Fig 6A and 6B). Namely, compared to the Baseline, the Monitoring-load and the Maintenance-load conditions showed increased connectivity within the VAN (p = .028 and p = .015, respectively), and between this network and the DMN (Monitoring: p = .018; Maintenance: p = .040). Additionally, in the Monitoring-load condition an increased InterNC was also found between the VAN and the DAN (p = .024). However, these results failed to reach significance after the FDR correction.

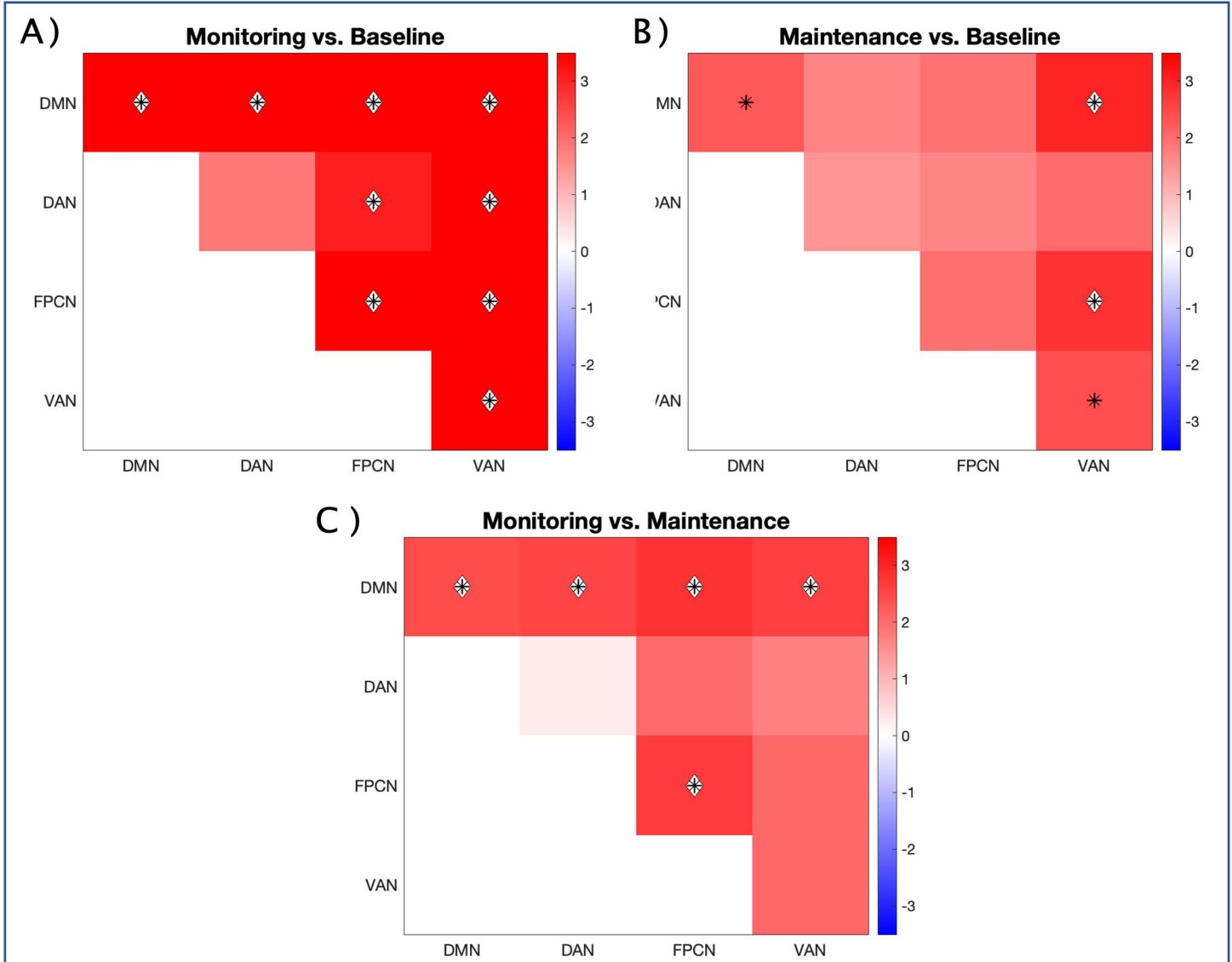

**Fig 5. Comparison of Functional Connectivity in the three conditions in the alpha band.** A. Comparison between the Monitoring-load and the Baseline conditions; B. Comparison between the Maintenance-load and the Baseline conditions; C. Comparison between the two PM conditions. Differences significant for p < .05 are displayed as black asterisks, whereas those significant for q < .05 are marked with a white diamond. The scale on the left represents t-values derived from statistical comparisons. DMN: Default Mode Network; DAN: Dorsal Attention Network; FPCN: FrontoParietal Control Network; VAN: Ventral Attention Network.

### Correlation between behavioral and FC measures

In general, all FC values—both InterNC and IntraNC—showed strong correlations with each other across all three conditions, both in the theta and alpha bands. Regarding their relationship with behavioral measures, significant correlations were found selectively in the alpha band, between FC measures and RTs (but not with accuracy). Notably, these correlations were found selectively in the two PM tasks, but not in the Baseline condition. In detail, InterNC and IntraNC values of the DAN and the VAN showed analogous associations with RTs in both PM conditions. However, the Monitoring-Load condition was characterized by a relationship between RT and InterNC of the FPCN (Rho = 0.466, p = 0.035), whereas in the

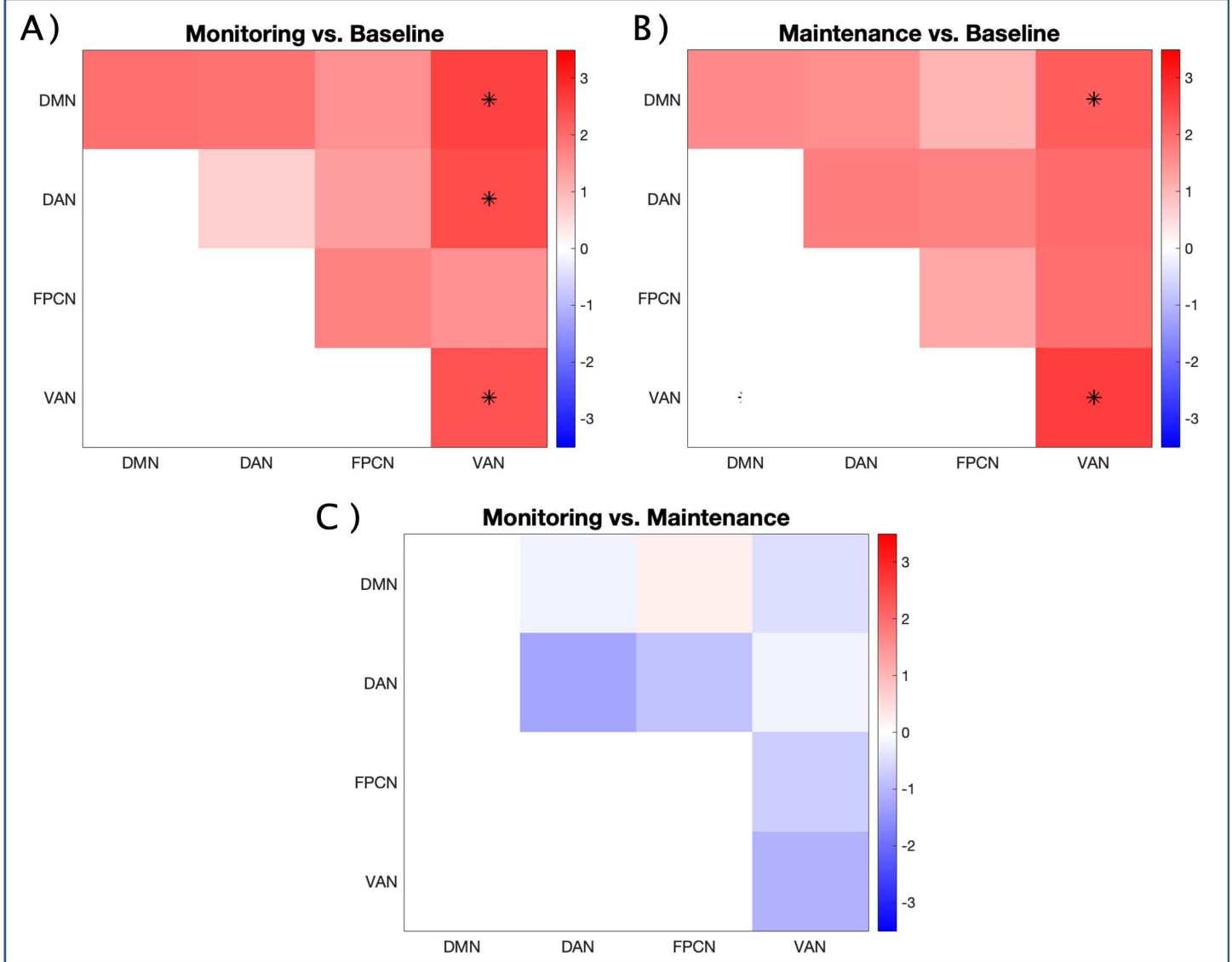

**Fig 6. Comparison of Functional Connectivity in the three conditions in the theta band.** A. Comparison between the Monitoring-load and the Baseline conditions; B. Comparison between the Maintenance-load and the Baseline conditions; **C.** Comparison between the two PM conditions. Differences significant for p < .05 are displayed as black asterisks, whereas those significant for q < .05 are marked with a white diamond. The scale on the left represents t-values derived from statistical comparisons. DMN: Default Mode Network; DAN: Dorsal Attention Network; FPCN: FrontoParietal Control Network; VAN: Ventral Attention Network.

Maintenance-Load condition RT resulted to correlate with IntraNC values of the DMN (Rho = 0.574, p = 0.007). Spearman's Rho and p-values for each significant correlation are displayed in Table 1, separately for network and condition.

## Comparison of power spectra

The repeated measures ANOVA (rmANOVA) on log-transformed alpha power revealed a significant main effect of Condition [$F_{(2, 40)} = 5.000$; $p = .012$; $\eta_p^2 = 0.200$] and a medium-sized interaction effect between Condition and Networks [$F_{(6, 120)} = 2.252$; $p = .043$; $\eta_p^2 = 0.101$] within the alpha band. Post-hoc analyses indicated that the primary

**Table 1. Connectivity-response times correlations.**

| | | Monitoring-Load | | Maintenance-Load | |
|---|---|---|---|---|---|
| | | **Rho** | **p-value** | **Rho** | **p-value** |
| **DAN** | InterNC | .506 | .020 | .458 | .038 |
| | IntraNC | .516 | .018 | .456 | .039 |
| **DMN** | InterNC | | | | |
| | IntraNC | | | .574 | .007 |
| **FPCN** | InterNC | .466 | .035 | | |
| | IntraNC | | | | |
| **VAN** | InterNC | .532 | .014 | .521 | .017 |
| | IntraNC | .475 | .031 | .510 | .019 |

Significant correlations between response times (RT) and connectivity measures (InterNC and IntraNC) for each network in the two PM conditions.

differences were between the Baseline and Monitoring-Load conditions ($p_{bonf}$ = .013), while the difference between the Baseline and Maintenance-Load conditions did not reach statistical significance ($p_{bonf}$ = .077). Comparing the results separately within each network, the only difference in log-transformed alpha power resulted to be the difference between the Baseline and the Monitoring-Load condition within the DAN ($p_{bonf}$ = .015). This result indicates that the DAN regions did not only display peculiar patterns of connectivity in the Monitoring-Load condition, but also enhanced levels of alpha power, a well-known marker of increased internal processing. The average alpha power of each network in the three conditions is displayed in Fig 7. Concerning the Theta band, no effects of Condition nor of its interaction with Network were found.

## Discussion

In this study, we examined functional connectivity (FC) patterns during the prestimulus time window across different PM conditions. In the Baseline condition, this interval corresponded to the preparation to perform the ongoing task (LDT) without additional instructions. Conversely, in the two PM conditions, it also corresponded to the monitoring/maintenance phase, during which participants were holding specific intentions in memory while monitoring the environment for the presentation of the associated stimuli. The two PM tasks varied in terms of their maintenance and monitoring demands, as in one condition participants were required to maintain multiple intentions in memory (Maintenance-load condition), whereas in the other they were required to detect less distinguishable PM cues (Monitoring-load condition). Our objective was to determine whether the enhancement of these cognitive processes was reflected in distinct patterns of connectivity within and between specific networks: the Dorsal Attention (DAN), the Default-Mode (DMN), the Frontoparietal Control (FPCN) and the Ventral Attention (VAN) networks. To our knowledge, this is the first study to apply FC analysis on neurophysiological data to investigate the neural bases of PM. Furthermore, the prestimulus phase of PM tasks has not previously been explored using neurophysiological techniques. This is because MEG and EEG analyses require the definition of a "baseline" time window, subtracting its activity from subsequent intervals of interest (typically corresponding to the period after stimulus presentation). However, we argue that the prestimulus window can be a fruitful source of information for anticipatory and preparatory cognitive processes involved in PM. Namely, in this interval it is possible to investigate the neural correlates of

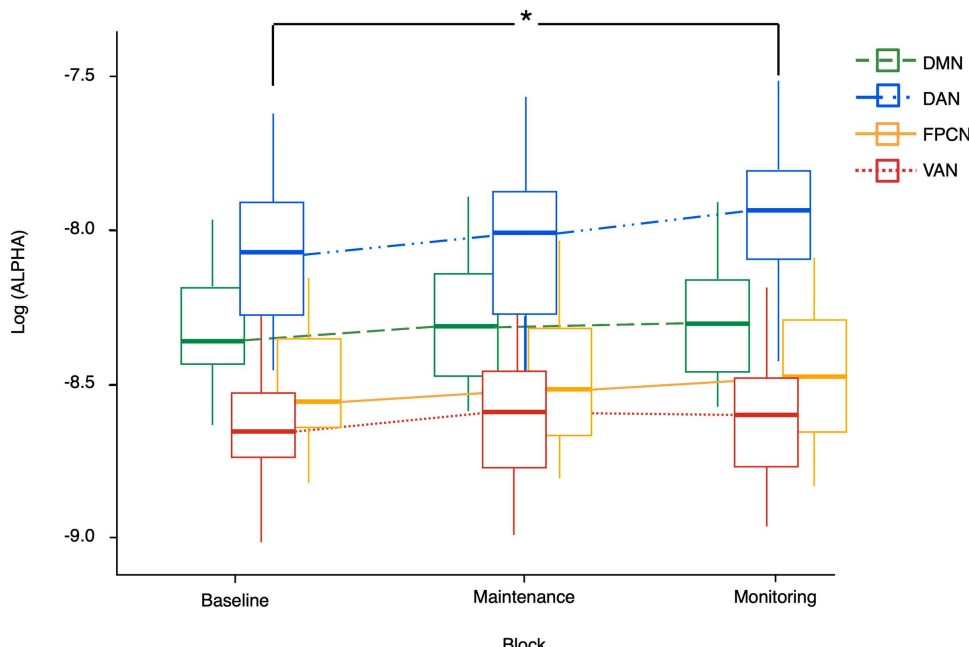

**Fig 7. Comparison of log-transformed power spectra in the alpha band.** Comparison of Alpha power for each each network in the three conditions. The asterisk represents significant differences between conditions (q < .05). DMN: Default Mode Network; DAN: Dorsal Attention Network; FPCN: FrontoParietal Control Network; VAN: Ventral Attention Network.

intention maintenance and strategic monitoring processes without possible confounding effects associated with the performance of the ongoing task.

FC analysis followed the approach proposed by Samogin et al. [37]. Namely, inter-network and intra-network connectivity values (InterNC and IntraNC, respectively) were calculated within and between each network of interest, separately for the alpha and theta band, then compared within and between each condition employing two complementary methods (see below). Subsequently, a correlation analysis was conducted to investigate the relationships between performance and these connectivity measures.

In general, both approaches identified FC differences within the alpha band, whereas the fewer differences observed in the theta band did not survive the correction for multiple comparisons. A similar difference between frequency bands was also observed in the correlational analysis. Arguably, the lack of significant differences in the theta band may be associated with the length of the time windows (as discussed in the Study Limitation section). Nonetheless, these results align with several studies indicating that the strongest correspondence between networks identified by fMRI (as the Yeo networks employed in this study) and by EEG/MEG occurs within the alpha band [66,67].

The first analysis consisted of the comparison between the InterNC and the IntraNC of each network, separately for conditions and frequency bands. This approach revealed the presence of networks that were significantly more connected with other networks than within themselves. In detail, the VAN showed patterns of significantly greater InterNC (compared to its IntraNC) across all conditions, suggesting a tendency of VAN regions to be significantly more connected with other networks than within themselves independently from the presence and type of PM task. Considering that the VAN is implicated in the automatic capture

of attention by salient stimuli [68], one plausible interpretation is that during the prestimulus interval, the VAN intensifies communication with the other networks to prepare for the detection and processing of the stimuli. In contrast, the DAN showed greater InterNC (compared to its IntraNC) selectively in the Monitoring-load condition. This observation aligns with the findings of several neuroimaging studies, that consistently reported increased activation of DAN regions during the monitoring/maintenance phase of PM tasks [69,70]. The selective presence of this pattern in the Monitoring-load condition provides strong support for the Attention to Delayed Intention (ATODI) model [12]. Specifically, given that the instruction of this PM task enhanced the reliance upon strategic monitoring, the increased connectivity of the DAN regions with other networks may reflect enhanced top-down monitoring processes that prepare the system to actively search for the PM cue. This interpretation is further supported by the results of the comparison of alpha power across conditions, which revealed that in the Monitoring-load condition the DAN exhibited not only increased InterNC but also enhanced alpha activity—a marker of heightened suppression of irrelevant information and enhanced top-down processing [71] (Fig 7).

The second analysis consisted of the comparison of FC across the three experimental conditions, both in terms of IntraNC and InterNC. Once again, significant findings were predominantly observed in the alpha band. These comparisons revealed different patterns of network connectivity depending on the requests of the PM tasks. Specifically, when comparing the Monitoring-load and the Baseline conditions a generalized increase in FC within and between all networks was observed, with the notable exception of the IntraNC of the DAN. The similar levels of IntraNC of the DAN between the Monitoring-load and the Baseline conditions integrate the previous findings by Cona and colleagues [23], suggesting that the greater activation previously found in DAN regions in the Monitoring-load condition (also replicated in the comparison of alpha power) primarily reflects enhanced communication with other networks, rather than within its own nodes. The Monitoring-load condition was also characterized by increased levels of IntraNC and InterNC in all the other networks (DMN, FPCN, and VAN). This finding may suggest that monitoring-enhancing PM instructions can induce a generalized "preparatory state" aimed at readiness to process forthcoming stimuli, in line with the Multiprocess Framework [7], which posits that PM task performance is supported by a dynamic interplay of automatic and controlled processes, depending on the nature of the task and environmental demands. This observation is also consistent with previous studies linking prestimulus alpha activity to top-down processes involved in preparing for task-performance [31,72]. The differences in connectivity patterns between the Maintenance-load and the Baseline conditions were more localized to specific networks. Notably, increased InterNC was observed between the VAN and regions of the DMN and the FPCN. The increased connectivity between the VAN and the DMN can be interpreted in the light of the PM task's demands, which enhanced maintenance load (promoting the focus on internal contents) while keeping monitoring requests low, thus allowing PM cue detection to rely on bottom-up processes. Critically, memory and internal processing have been consistently associated with the enrollment of DMN regions [32,33], whereas the elaboration of salient stimuli is linked to bottom-up mechanisms relying on VAN regions [20,21]. Similarly, the increased InterNC between the VAN and FPCN can be explained by the FPCN role in regulating and integrating information between other networks, such as the VAN and the DMN [42].

The comparison between the two PM conditions revealed that connectivity in the Monitoring-load condition was significantly higher within regions of the FPCN and within regions of the DMN. Additionally, increased connectivity was found between the DMN and all other networks, including the DAN and the VAN. The increased involvement of the DMN in the monitoring-enhancing condition may seem counterintuitive. However, considering that the analyses focused

on the prestimulus time window, a possible interpretation is that in the Monitoring-load condition top-down processes were allocated before the stimulus presentation to prepare for intention retrieval (strategic monitoring). In contrast, in the Maintenance-load condition, the processes underlying intention retrieval occurred only after the detection of the PM cue, to retrieve the correct intention—among multiple ones—related to that cue. In line with this hypothesis, the DMN increased its connectivity with the FPCN during internally directed complex processes such as goal-directed cognition and autobiographical planning [73,74]. Therefore, the differences between the two PM conditions can reflect the engagement in strategic monitoring processes, which may also be oriented toward internal contents (the intention and the associated representation of the PM cue) during the maintenance/monitoring phase.

Finally, the correlation analysis revealed significant associations between Response Times (RTs) and prestimulus alpha connectivity values. Specifically, both InterNC and IntraNC of certain networks positively correlated with RTs, consistent with prior studies reporting increased prestimulus alpha power in trials with longer response times [72,75]. Notably, these correlations were observed exclusively in the two prospective conditions (and not in the Baseline) suggesting that they may reflect task-specific preparatory mechanisms. In detail, the DAN and VAN consistently showed positive correlations with RTs across both PM tasks. In the Monitoring-Load condition, RTs also correlated with the InterNC of the FPCN, whereas the Maintenance-Load condition was characterized by a correlation between RTs and IntraNC within DMN regions. Considering the functional roles of the two networks—top-down control and communication among other networks for the FPCN and internal content processing for the DMN—the selective positive correlations of these two connectivity measures with RTs in the two prospective conditions (enhancing monitoring or maintenance demands, respectively) support the hypothesis of a dynamic interplay between networks in adapting to varying prospective task demands.

Altogether, these results indicate that the modulation of monitoring and maintenance requests can affect the mechanisms and networks involved in prospective remembering, even before the stimulus presentation. PM tasks promoting monitoring processes (i.e., strategic monitoring) induce a preparatory state for the detection of PM cues, reflected in a generalized strengthening of connectivity across all the networks. In this interval, the connectivity of the DAN regions is significantly more directed outward, to enhance top-down organization and communication with the other networks. In the Maintenance-load condition, on the other hand, the combination of enhanced memory load (through multiple intentions to maintain) and the salience of the associated PM cue (which reduced the attentional load) induces more selective increases in connectivity. These increases occur between the network responsible for bottom-up capture of attention (VAN) and the networks involved in the processing of internal contents (DMN and FPCN). Finally, the comparison between the Monitoring-Load and the Maintenance-Load PM tasks revealed increased connectivity within the DMN and FPCN, and between DMN and all the other networks, in the former. We believe this finding reflects enhanced internal goal-directed processing in the prestimulus interval of this PM condition, a process that in the Maintenance-load condition occurs only after the detection of the PM cue.

## Study limitations and future perspectives

This study has some limitations that must be considered. The first limitation to address consist in the length of the epochs considered for the analysis. Namely, connectivity was investigated in the 1000 ms preceding the presentation of the stimulus. Generally, intervals comprising at least three full cycles of a specific frequency band are recommended to perform FC analysis [65]. While the duration of one second is consistent with this

recommendation for both alpha (8–12 Hz, meaning on average ten full cycles per second) and theta (5–7 Hz, six full cycles per second) bands, the failure to identify significant differences in the latter may indicate that the considered time window was too short to be sensitive to the differences between conditions. Methodological studies exploring the effect of epoch length on EEG functional connectivity have shown that longer time windows produce more reliable results [76]. A second limitation consists in the impossibility to analyze post-stimulus functional connectivity and compare it with prestimulus findings. Namely, behavioral responses were given as soon as 400 ms after stimulus presentation, thus not allowing to perform connectivity analysis in the post-stimulus time window to compare it with the prestimulus one. Future paradigms should use longer prestimulus time windows to increase the reliability of findings on IntraNC and InterNC in the theta band, and consider the implementation of delayed response design in the experimental tasks (i.e., asking participants to wait for the appearance of a second signal before responding to each trial) to enable the comparison between prestimulus and post-stimulus connectivity patterns. This procedure would enhance the reliability of the results and provide clearer reference to the time-frequency and neuroimaging findings collected during the maintenance/monitoring and the retrieval phases of PM. Additionally, future research should also explore the directionality of these interactions in the frequency domain, which may provide further insight into the mechanisms underlying PM tasks, employing approaches such as the directed transfer function, which has been effectively applied to MEG data at the source level [77]. Investigating the directional flow of information between neural regions might deepen our understanding of communication dynamics during PM processing.

Despite these limitations, the analyses conducted in this study provided evidence of significant differences in connectivity arising before stimulus presentation (maintenance/monitoring phase of PM) depending on the PM task demands. Hopefully, these findings will stimulate future research to broaden the investigation of this phenomenon under various conditions. Furthermore, considering that the flexibility of FC approaches allows the application of this type of analysis to continuous (not stimulus-locked) data, the current study might inspire researchers to conduct PM investigation using more ecological paradigms, involving simpler and less demanding ongoing tasks, and employing this methodology to explore the mechanisms underlying intention maintenance in more comprised clinical populations.

## Conclusions

This study shed light on the mechanisms and brain networks underlying the maintenance of an intention in memory and the monitoring strategies to detect the related PM cue, highlighting differences associated with either attentional-enhancing or memory-enhancing features of the PM task. Heightened monitoring demands were associated with increased connectivity of the Dorsal Attention Network (DAN) with other networks rather than within itself, alongside a general state of increased connectivity across all networks considered. Conversely, increased maintenance load was characterized by more targeted increases in connectivity, particularly between networks involved in internal processing (Default Mode Network, DMN, and Frontoparietal Control Network, FPCN) and those supporting bottom-up attentional processes (Ventral Attention Network, VAN). In conclusion, the employment of a functional connectivity approach was found to have substantial potential for investigating PM, supporting the hypothesis that multiple pathways (and mechanisms) are involved in prospective remembering, increasing the knowledge of the specific networks involved in PM depending on the features of the prospective task.

## Supporting information

**S1 Table. Correspondence between Destrieux regions and Yeo Networks.** Table displaying correspondences between each Region of Interest (Destrieux Atlas) and the corresponding Yeo Network. Coordinates indicate the centroid of each region.
(DOCX)

**S1 Fig. Comparison between InterNC and IntraNC for the seven Yeo networks in each condition.** Networks presenting significant differences between InterNC and IntraNC in the alpha band. Differences significant for p < .05 are displayed as black asterisks, whereas those significant for q < .05 are marked with a white diamond. The scale on the left represents t-values derived from statistical comparisons. DMN: Default Mode Network; DAN: Dorsal Attention Network; FPCN: FrontoParietal Control Network; LN: Limbic Network; SMN: SomatoMotor Network; VAN: Ventral Attention Network; VN: Visual Network.
(TIF)

**S2 Fig. Comparison of InterNC and IntraNC for the seven Yeo network across the three conditions.** Comparison of Functional Connectivity in the three conditions in the alpha band. A. Comparison between the Monitoring-load and the Baseline conditions; B. Comparison between the Maintenance-load and the Baseline conditions; C. Comparison between the two PM conditions. Differences significant for p < .05 are displayed as black asterisks, whereas those significant for q < .05 are marked with a white diamond. The scale on the left represents t-values derived from statistical comparisons. DMN: Default Mode Network; DAN: Dorsal Attention Network; FPCN: FrontoParietal Control Network; LN: Limbic Network; SMN: SomatoMotor Network; VAN: Ventral Attention Network; VN: Visual Network.
(TIF)

## Author contributions

**Conceptualization:** Giorgia Cona, Patrizia Bisiacchi, Giorgio Arcara.

**Data curation:** Giorgio Arcara.

**Formal analysis:** Stefano Vicentin, Dante Mantini.

**Investigation:** Giorgia Cona.

**Methodology:** Stefano Vicentin, Marco Marino, Dante Mantini, Giorgio Arcara.

**Project administration:** Giorgia Cona, Patrizia Bisiacchi, Dante Mantini, Giorgio Arcara.

**Resources:** Giorgio Arcara.

**Software:** Marco Marino.

**Supervision:** Giorgia Cona, Patrizia Bisiacchi, Dante Mantini.

**Validation:** Giorgio Arcara.

**Visualization:** Marco Marino, Giorgio Arcara.

**Writing – original draft:** Stefano Vicentin.

**Writing – review & editing:** Patrizia Bisiacchi, Dante Mantini, Giorgio Arcara.

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
