## [Decision Letter · Decision Letter 0]

2 Dec 2024

PONE-D-24-30533Prestimulus functional connectivity reflects attention orientation in a prospective memory task: a magnetoencephalographic (MEG) studyPLOS ONE

Dear Dr. VICENTIN,

Thank you for submitting your manuscript to PLOS ONE. After careful consideration, we feel that it has merit but does not fully meet PLOS ONE’s publication criteria as it currently stands. Therefore, we invite you to submit a revised version of the manuscript that addresses the points raised during the review process.

We look forward to receiving your revised manuscript.

Kind regards,

Farzin Hajebrahimi, Ph.D

Academic Editor

PLOS ONE

Journal Requirements:

2. Please note that in order to use the direct billing option the corresponding author must be affiliated with the chosen institute. Please either amend your manuscript to change the affiliation or corresponding author, or email us at plosone@plos.org with a request to remove this option.

4. Please remove your figures from within your manuscript file, leaving only the individual TIFF/EPS image files, uploaded separately. These will be automatically included in the reviewers’ PDF**.**

**5. ** We note that Figure 1 in your submission contain copyrighted images. All PLOS content is published under the Creative Commons Attribution License (CC BY 4.0), which means that the manuscript, images, and Supporting Information files will be freely available online, and any third party is permitted to access, download, copy, distribute, and use these materials in any way, even commercially, with proper attribution. For more information, see our copyright guidelines: http://journals.plos.org/plosone/s/licenses-and-copyright. We require you to either (1) present written permission from the copyright holder to publish these figures specifically under the CC BY 4.0 license, or (2) remove the figures from your submission:

a. You may seek permission from the original copyright holder of Figure 1 to publish the content specifically under the CC BY 4.0 license. We recommend that you contact the original copyright holder with the Content Permission Form (http://journals.plos.org/plosone/s/file?id=7c09/content-permission-form.pdf) and the following text: “I request permission for the open-access journal PLOS ONE to publish XXX under the Creative Commons Attribution License (CCAL) CC BY 4.0 (http://creativecommons.org/licenses/by/4.0/). Please be aware that this license allows unrestricted use and distribution, even commercially, by third parties. Please reply and provide explicit written permission to publish XXX under a CC BY license and complete the attached form.” Please upload the completed Content Permission Form or other proof of granted permissions as an "Other" file with your submission. In the figure caption of the copyrighted figure, please include the following text: “Reprinted from [ref] under a CC BY license, with permission from [name of publisher], original copyright [original copyright year].”

6. Please include captions for your Supporting Information files at the end of your manuscript, and update any in-text citations to match accordingly. Please see our Supporting Information guidelines for more information: http://journals.plos.org/plosone/s/supporting-information .

Additional Editor Comments:

Please provide details of the data acquisition parameters. For Freesurfer, please explain details of the preprocessing.

For the whole manuscript, please be consistent about within- and between-network or inter/intra.

Reviewers' comments:

Reviewer's Responses to Questions

**Comments to the Author**

1. Is the manuscript technically sound, and do the data support the conclusions?

Reviewer #1: No

Reviewer #2: Yes

Reviewer #3: Yes

2. Has the statistical analysis been performed appropriately and rigorously? 

Reviewer #1: No

Reviewer #2: Yes

Reviewer #3: Yes

3. Have the authors made all data underlying the findings in their manuscript fully available?

The PLOS Data policy requires authors to make all data underlying the findings described in their manuscript fully available without restriction, with rare exception (please refer to the Data Availability Statement in the manuscript PDF file). The data should be provided as part of the manuscript or its supporting information, or deposited to a public repository. For example, in addition to summary statistics, the data points behind means, medians and variance measures should be available. If there are restrictions on publicly sharing data—e.g. participant privacy or use of data from a third party—those must be specified

Reviewer #1: Yes

Reviewer #2: Yes

Reviewer #3: Yes

4. Is the manuscript presented in an intelligible fashion and written in standard English?

Reviewer #1: Yes

Reviewer #2: Yes

Reviewer #3: Yes

5. Review Comments to the Author

Reviewer #1: Review comments:

This study investigates prestimulus functional connectivity (FC) related to attention orientation in a prospective memory (PM) task using MEG. The authors differentiated the FC patterns between the monitoring and maintenance PM tasks during the prestimulus period. They observed increased inter-network FC of the dorsal attention network in the alpha band for the monitoring-load task, and increased FC of the ventral attention network with the frontoparietal and default mode networks for the maintenance-load task. These results demonstrate that different networks are engaged in PM depending on the specific features of the task.

While I find this manuscript interesting, I have serious concerns about the analysis and the results. I hope the authors can address and clarify some of the ambiguities present in the current manuscript.

1. The authors introduce three models/theories: the preparatory attention and memory theory, the multiprocess framework, and the attention to the delayed intention model. These theories offer different interpretations of the mechanisms underlying PM. It would be helpful if the authors could clarify whether the FC data from the current study supports any of these theoretical accounts. Providing an integrated view that links the neural data with these theoretical frameworks could strengthen the manuscript, as the relationship between the cognitive theories and neural data is currently unclear.

2. While I agree that MEG offers excellent temporal resolution compared to fMRI, I feel that this advantage could be better shown in the study. Could the authors provide more detailed temporal analyses of FC during the prestimulus interval? This could highlight the unique temporal dynamics captured by MEG.

3. The task description is somewhat unclear. To help readers better understand the experimental design and procedure, it would be beneficial if the authors provided visual illustrations, including timing and trial examples, in a figure.

4. Behavioural results are also crucial for understanding how the prestimulus neural data relates to task performance. Could the authors include these results in the manuscript?

5. It is unclear which brain hemisphere's data were analysed for the FC analysis. Clarifying this would help readers understand the lateralization of the observed effects.

6. I have some questions regarding the FC analysis method. Could the authors clarify whether they extracted and normalized the correlation values before averaging and comparing the IntraNC and InterNC values among the conditions? It is important to ensure that any averaging of correlation values is done following proper normalization, as correlation is not a linear measure and cannot be simply averaged without introducing statistical inaccuracies.

7. Following up on the previous comment, could the authors elaborate on the statistical analysis conducted based on raw or normalized power values in Figure 7 (or on p.22)? I am uncertain whether statistical tests can be appropriately conducted on raw power spectra. A more detailed explanation would enhance the clarity of the results.

8. Could the authors specify the scale used in Figures 3 through 6? This would aid in the interpretation of the data presented.

Reviewer #2: This study investigates the neural bases of prospective memory (PM) from a network connectivity perspective, using MEG data to assess functional connectivity (FC) across different brain networks under varying PM task demands. Specifically, the authors analyze two PM conditions—monitoring-load and maintenance-load—that emphasize either attentional monitoring or memory maintenance. By focusing on the prestimulus phase, the study identifies distinct connectivity patterns: increased inter-network FC of the Dorsal Attention Network (DAN) in the alpha band in the monitoring-load condition, and heightened connectivity of the Ventral Attention Network (VAN) with the FrontoParietal Control and Default Mode Networks (FPCN and DMN) in the maintenance-load condition. These findings suggest that top-down and bottom-up mechanisms are differentially engaged depending on PM task requirements.

While the study addresses an important and innovative approach to understanding PM processes through functional connectivity, several points require clarification and further elaboration to strengthen the manuscript.

Introduction

Page 3: Although original, the anecdote about Amedeo Modigliani’s paintings does not seem entirely appropriate for introducing the concept of prospective memory (PM) in a scientific context. I suggest reserving this example for a more popular science publication and omitting it from the scientific paper, as it may detract from the formality and focus of the study.

Page 4, line 82: In the retrieval phase, it should be noted that the associated intention must not only be correctly retrieved but also executed. Including an example to explain each phase would improve clarity.

Page 5: When discussing The Multiprocess Framework, it’s important to address focal and non-focal cues and provide clear explanations, particularly since focality and salience are features manipulated in this experiment.

Page 5: “However, opposite patterns of activation have been observed in the medial and lateral aPFC and between the ventral and dorsal portions of the FPN during specific PM phases or in relation to PM task features.” This needs a brief explanation of how task features impact activation in these regions. Include a review of previous studies that have examined the role of the number of intentions, salience, and focality in PM, focusing on studies that explore these aspects at a neural level.

Page 8. I recommend including a section titled "In the Present Study" to improve the flow and clarity of the text. In the hypotheses, clearly state the expected behavioral outcomes as well.

Materials and Methods

Page 10. Data Acquisition: Please specify any inclusion or exclusion criteria for participant recruitment and indicate whether participants received any form of compensation. Additionally, clarify how the sample size was determined. Conducting a power analysis to estimate the expected effect size would further support the adequacy of the chosen sample size.

Page 10. For MRI data, provide additional imaging parameters such as voxel size, repetition time, and echo time.

Page 10-11. Task procedure: It is essential to clarify certain aspects of the experimental setup to improve transparency and reproducibility. What were the specific timings and intervals for each trial and block, including the presentation duration of task stimuli, inter-trial intervals, number of PM and ongoing trials per block, and any breaks between blocks? What mode and device were used to present stimuli, and what software was employed for stimulus delivery and response recording? Additionally, how were responses collected, and was response latency recorded? In the Maintenance-load block participants were instructed to maintain three different intentions in memory—what specifically were these intentions? Finally, did participants receive any practice or familiarization trials to ensure understanding of the tasks before starting the experimental blocks?

Page 12.” …considering only the ongoing trials with a correct response.” Please provide the average number of ongoing trials included per participant and block.

Page 12. Mention if any power analysis was conducted in the different analyses to ensure sufficient sensitivity for the paired t-tests, particularly given the multiple comparisons and FDR correction.

Page 13-14. Please 1) include specific hypotheses regarding the IntraNC and InterNC comparisons within the hypotheses section of the experiment, and 2) explain how these processes of segregation and integration relate to PM processes within each of the networks of interest. This addition will clarify the expected outcomes and their connection to PM mechanisms across different networks.

Results

Page 17. What about the behavioral results? Please include them to confirm that the typical effects of a PM task are observed. Additionally, conducting a correlation analysis between behavioral outcomes and neuroimaging data could help interpret how functional connectivity influences performance in a prospective memory task. This would provide a more comprehensive understanding of the relationship between brain connectivity and behavioral execution in PM contexts.

Page 17. Include a brief discussion on the functional significance of these findings within each subsection, even if further elaborated in the discussion. This will provide readers with immediate insight into the relevance of the observed changes in the context of PM tasks.

Page 22. Specify in the hypotheses section whether increases in alpha or theta power were expected in certain networks across conditions, based on prior literature or theoretical models of PM.

Discussion

Page 23. How can you ensure that the prestimulus phase is not influenced by performance in the immediately preceding trial? Could there be differences in this phase resulting from a correct or incorrect response in the preceding ongoing trial? How might this be controlled?

Page 23. The PM cues in the different conditions (maintenance vs.monitoring) differ in focality and salience. While salience refers to the ease or difficulty with which a PM cue is perceptually detected, focality pertains to the similarity in processing between the PM cue stimulus and the stimulus in ongoing trials. Although both characteristics relate to detection and monitoring processes in PM, could each affect functional connectivity of these processes differently? Additionally, how might the results change if only focality were manipulated, given its direct connection to different retrieval mechanisms within the Multiprocess Framework?

Page 26. "In contrast, in the Maintenance-load condition, the processes underlying intention retrieval occurred only after the detection of the PM cue, to retrieve the correct intention among multiple ones related to that cue"— I agree with this interpretation. To support this claim, is it possible to perform these post-stimulus analyses with the data you have?

Study Limitations and Future Perspectives

Page 28. There seems to be a potential contradiction here. In page 12, you mention segmenting the signal into epochs from 1.5 seconds before to 1.5 seconds after stimulus presentation, which would suggest that post-stimulus connectivity analysis is possible. However, in the limitations section, you note that post-stimulus analysis is not feasible due to participants' rapid responses (as early as 400 ms). Could you clarify this point? Specifically, is it possible to conduct any preliminary post-stimulus analysis within the available 1.5-second epoch, even if limited?

Reviewer #3: In this paper, authors investigated the effects of external (monitoring condition) and internal (maintenance condition) attention to detect a Prospective Memory cue on functional connectivity using MEG data. The neurosphysiological data demonstrated that the monitoring-load condition with respect to the baseline condition was characterized by increased inter-network FC of the DAN and that the maintenance-load condition (with respect to the baseline condition) was characterized by increased inter-network connectivity between the VAN and the FPN and the DMN. Both the modulations are observed in the alpha band. In addition, the authors hypothesized specific modulations in the theta band that however failed to reach significance after the correction of multiple comparisons for probable methodological concerns (i.e. the length of the prestimulus time window). The aim of the work is clear and well written.

Comments:

I understand why the authors focus the analysis on the networks involved in PM processing (i.e. DAN, DMN and VAN) and on the FPCN. However I believe that showing functional connectivity also within and between the other Yeo networks (not employed in this study) could strengthen the specificity of the results. If the authors believe that this whole-brain analysis would worsen the clarity of the text, they could add findings in supplementary materials.

In addition, I think that it would be very interesting to evaluate the directionality of these interactions in the frequency domain, for example using the directed transfer function that has been used on MEG data at the source level (e.g. using the Atlantis Processing Toolbox: atlantis.psychologia.uj.edu.pl, Spadone et al., 2021 Brain Connectivity). The authors could investigate this in future research and discuss the importance of examining directionality in communication.

6. PLOS authors have the option to publish the peer review history of their article (what does this mean? ). If published, this will include your full peer review and any attached files.

**Do you want your identity to be public for this peer review?** For information about this choice, including consent withdrawal, please see our Privacy Policy .

Reviewer #1: No

Reviewer #2: No

Reviewer #3: No

---

## [Author Response · Author response to Decision Letter 1]

11 Jan 2025

Academic Editor: Please ensure that your manuscript meets PLOS ONE's style requirements, including those for file naming. The PLOS ONE style templates can be found at

2. Please note that in order to use the direct billing option the corresponding author must be affiliated with the chosen institute. Please either amend your manuscript to change the affiliation or corresponding author, or email us at plosone@plos.org with a request to remove this option.

Reply: We thank the Editor for the indication and included the necessary information in the Funding Information Section. The Institution that will fund the Open Access publication is the same where the Corresponding Author works, i.e., the University of Padova.

Reply: According to the Editor request, we included the information associated with the ethics committee and the study approvation number for the study in the Methods Section.

Changes made to the manuscript (Page 10): “The study was conducted following the guidelines of the Helsinki Declaration and received approval from the local ethics committee (Etichs Committee for clinical sperimentation of Venice, approval n. 1013/IRCCSsancamillo). Exclusion criteria included history of epileptic seizures, neurological or psychiatric disorders, metal implants in their body, acoustic implants, non-removable piercings or metal accessories. Participants took part volountarly to the study and did not receive any form of compensation.”

4. Please remove your figures from within your manuscript file, leaving only the individual TIFF/EPS image files, uploaded separately. These will be automatically included in the reviewers’ PDF.

Reply: We apologize for the mistake of including the images in the manuscript. According to the PLOS one guidelines, images are now included only as separate files. Each image is referred in the manuscript with the indication [Figure N.] to indicate the place the image should be placed.

5. We note that Figure 1 in your submission contain copyrighted images. All PLOS content is published under the Creative Commons Attribution License (CC BY 4.0), which means that the manuscript, images, and Supporting Information files will be freely available online, and any third party is permitted to access, download, copy, distribute, and use these materials in any way, even commercially, with proper attribution. For more information, see our copyright guidelines: http://journals.plos.org/plosone/s/licenses-and-copyright. We require you to either (1) present written permission from the copyright holder to publish these figures specifically under the CC BY 4.0 license, or (2) remove the figures from your submission.

Reply: The painting displayed in Figure 1 is distributed by the Metropolitan Museum of Art for unrestricted use under Creative Commons Zero (CC0), allowing to share, modify and publish this image with no restrictions. However, following the suggestion of Reviewer 2, we decided to remove the figure from the manuscript.

Reply: According to the Editor’s suggestion, we included a caption for the Supporting Information file at the end of the manuscript.

Changes made to the manuscript (Page 42): “S1 Table. Correspondence between Destrieux regions and Yeo Networks. Table displaying correspondences between each Region of Interest (Destrieux Atlas) and the corresponding Yeo Netwok. Coordinates indicate the centroid of each region.

S1 Fig. Comparison between InterNC and IntraNC for the seven Yeo networks in each condition.

Networks presenting significant differences between InterNC and IntraNC in the alpha band. Differences significant for p < .05 are displayed as black asterisks, whereas those significant for q < .05 are marked with a white diamond. The scale on the left represents t-values derived from statistical comparisons. DMN: Default Mode Network; DAN: Dorsal Attention Network; FPCN: FrontoParietal Control Network; LN: Limbic Network; SMN: SomatoMotor Network; VAN: Ventral Attention Network; VN: Visual Network.

S2 Fig. Comparison of InterNC and IntraNC for the seven Yeo network across the three conditions. Comparison of Functional Connectivity in the three conditions in the alpha band. A. Comparison between the Monitoring-load and the Baseline conditions; B. Comparison between the Maintenance-load and the Baseline conditions; C. Comparison between the two PM conditions. Differences significant for p < .05 are displayed as black asterisks, whereas those significant for q < .05 are marked with a white diamond. The scale on the left represents t-values derived from statistical comparisons. DMN: Default Mode Network; DAN: Dorsal Attention Network; FPCN: FrontoParietal Control Network; LN: Limbic Network; SMN: SomatoMotor Network; VAN: Ventral Attention Network; VN: Visual Network.”

Additional Editor Comments: Please provide details of the data acquisition parameters. For Freesurfer, please explain details of the preprocessing. For the whole manuscript, please be consistent about within- and between-network or inter/intra.

Reply: We thank the Editor for pointing out these issues. Regarding the preprocessing, we re-organized the information on MEG preprocessing, making them clearer and easier to follow, and included some missing information about MRI preprocessing. Regarding the terms “within” and “between”, they were replaced with the terms “intra” and “inter” throughout the whole manuscript to increase the consistency across the manuscript.

Changes made to the manuscript (Page 10-11): “The MEG preparation session involved the placement of three head coils at anatomical landmarks (left and right preauricular points, nasion) for head localization and co-registration with anatomical images. Additionally, six amagnetic electrodes were used for auxiliary recordings: four were positioned around the eyes (two for detecting vertical eye movements and blinks, and two for horizontal eye movements), and two were placed on the left and right scapulas to capture cardiac activity. The positions of the head coils and the participant’s head shape were digitized using a Polhemus Fastrak system to enable precise co-registration. After the preparation, participants were seated comfortably inside a Faraday cage and given instructions about the experimental tasks. They were informed that they would be monitored and could communicate at any time during the session. Participants were also assured they could request a break or terminate the experiment if they felt uncomfortable in the MEG room. During data acquisition, MEG signals were recorded continuously at a sampling rate of 1200 Hz, with a hardware anti-aliasing low-pass filter applied at 300 Hz.

In the second session, neuroimaging data were collected using a 1.5 Tesla Philips Achieva MRI scanner. Specifically, a T1-weighted whole-head anatomical image was acquired for each participant (repetition time [TR]: 8.3 ms, echo time [TE]: 4.1 ms, flip angle: 8°, acquired matrix resolution [MR] = 288 × 288, slice thickness [ST] = 0.87 mm). These images were used to create individual head models for MEG source reconstruction. Preprocessing of MRI data was performed using FreeSurfer [44] and included skull stripping, segmentation of subcortical structures, cortical surface reconstruction, and the generation of smoothed pial and white matter surfaces.

Reviewer #1: This study investigates prestimulus functional connectivity (FC) related to attention orientation in a prospective memory (PM) task using MEG. The authors differentiated the FC patterns between the monitoring and maintenance PM tasks during the prestimulus period. They observed increased inter-network FC of the dorsal attention network in the alpha band for the monitoring-load task, and increased FC of the ventral attention network with the frontoparietal and default mode networks for the maintenance-load task. These results demonstrate that different networks are engaged in PM depending on the specific features of the task. While I find this manuscript interesting, I have serious concerns about the analysis and the results. I hope the authors can address and clarify some of the ambiguities present in the current manuscript.

1. The authors introduce three models/theories: the preparatory attention and memory theory, the multiprocess framework, and the attention to the delayed intention model. These theories offer different interpretations of the mechanisms underlying PM. It would be helpful if the authors could clarify whether the FC data from the current study supports any of these theoretical accounts. Providing an integrated view that links the neural data with these theoretical frameworks could strengthen the manuscript, as the relationship between the cognitive theories and neural data is currently unclear.

Reply: We sincerely thank the Reviewer for this insightful comment. In the Discussion section of the manuscript, we addressed the interpretability of our functional connectivity (FC) findings within the context of the theoretical frameworks presented. Specifically, we discussed how our results could be interpreted as supporting the Attention to Delayed Intention (ATODI) model (see page 25) and the Multiprocess Framework (see page 26). To address the Reviewer’s concerns and improve clarity, we have expanded the Discussion section to include more explicit links between the FC results and these cognitive theories. We believe these additions not only clarify the connection between our findings and the theoretical frameworks but also strengthen the overall interpretability of our results. We trust that these revisions address the Reviewer’s concerns, but we are happy to make further modifications if needed.

Changes made to the manuscript (Page 25): “The selective presence of this pattern in the Monitoring-load condition provides strong support for the Attention to Delayed Intention (ATODI) model [12]. Specifically, given that the instruction of this PM task enhanced the reliance upon strategic monitoring, the increased connectivity of the DAN regions with other networks may reflect enhanced top-down monitoring processes that prepare the system to actively search for the PM cue. This interpretation is further supported by the results of the comparison of alpha power across conditions, which revealed that in the Monitoring-load condition the DAN exhibited not only increased InterNC but also enhanced alpha activity—a marker of heightened suppression of irrelevant information and enhanced top-down processing [72] (Fig 7).”

Changes made to the manuscript (Page 26): “This finding may suggest that monitoring-enhancing PM instructions can induce a generalized "preparatory state" aimed at readiness to process forthcoming stimuli, in line with the Multiprocess Framework [7], which posits that PM task performance is supported by a dynamic interplay of automatic and controlled processes, depending on the nature of the task and environmental demands.”

2. While I agree that MEG offers excellent temporal resolution compared to fMRI, I feel that this advantage could be better shown in the study. Could the authors provide more detailed temporal analyses of FC during the prestimulus interval? This could highlight the unique temporal dynamics captured by MEG.

Reply: We appreciate the Reviewer’s suggestion to conduct more time-locked analyses within the prestimulus interval to better showcase MEG's temporal precision. While we agree that additional approaches could have further highlighted the temporal dynamics of MEG, several theoretical considerations guided our decision not to pursue this in the current study. First, the inter-trial interval (ITI) was jittered to maintain participant vigilance and reduce similarities in processes across prestimulus intervals. This variability in timing complicates the use of precise time-locked analyses. Second, as discussed in the manuscript, defining a baseline interval to subtract average neural activity is a standard step for many analyses. While baseline normalization is not strictly mandatory, its absence is often discouraged in the literature. Without a clear stimulus to anchor epochs or define a baseline, time-locked analyses would have introduced interpretive challenges.

Instead, we focused on functional connectivity (FC) analyses, which are well-suited for investigating prestimulus activity without requiring time-locking to a specific event or baseline correction. Additionally, the unique temporal resolution of MEG allowed us to disentangle neural signals across distinct frequency bands (alpha, theta) and to analyze the patterns of connectivity within these bands, thereby leveraging MEG's strengths in capturing frequency-specific dynamics. We believe this approach aligns with the study's goals while adhering to methodological best practices.

3. The task description is somewhat unclear. To help readers better understand the experimental design and procedure, it would be beneficial if the authors provided visual illustrations, including timing and trial examples, in a figure.

Reply: We thank the Reviewer for this valuable suggestion. We agree that the original description of the experimental paradigm lacked clarity. In response, we have expanded the Task Procedure section to provide a more detailed account of the experimental design. Additionally, we have included a visual illustration of the paradigm to further enhance clarity and facilitate reader understanding. This illustration has been uploaded as Figure 1, and its corresponding caption has been incorporated into the manuscript.

Changes made to the manuscript (Page 13): “Fig 1. General structure of the experimental paradigm. Independently from the condition, the strings of letters for the LDT were presented until a response was given, or for a maximum of 3000 ms. Hence, a blank screen was presented for 1000 ms, followed by a fixation cross displayed for an interval of 1500 ± 250 ms. A) In the Baseline condition, participants only had to perform the LDT. B) The structure of the Monitoring-Load condition was identical to the one of the Baseline, except for the addition of the instruction to press a different button when the syllable “PRA” was detected. C) In the Memory-Load condition, participants were instructed to press three different buttons at the presentation of three underlined words (AZZURRO, GRIGIO, and ROSSO).”

4. Behavioural results are also crucial for understanding how the prestimulus neural data relates to task performance. Could the authors include these results in the manuscript?

Reply: We appreciate the Reviewer’s thoughtful comment regarding the importance of behavioral results for understanding how prestimulus connectivity relates to performance. The behavioral results for this dataset were analyzed and discussed in a previous publication, which we have referenced throughout the manuscript. To avoid redundancy, we did not analyze or discuss those results in this study. However, following a suggestion from Reviewer 2, we analyzed the relationships between behavioral data and prestimulus connectivity values. This new analysis revealed intriguing differences across conditions, offering further insights into the interplay between functional connectivity (FC) patterns and task performance. Specifically, we found that reaction times (RTs) were significantly correlated with FC values in the prestimulus alpha band during the two prospective memory (PM) tasks (Monitoring-Loa

---

## [Decision Letter · Decision Letter 1]

29 Jan 2025

Prestimulus functional connectivity reflects attention orientation in a prospective memory task: a magnetoencephalographic (MEG) study

PONE-D-24-30533R1

Dear Dr. VICENTIN,

We’re pleased to inform you that your manuscript has been judged scientifically suitable for publication and will be formally accepted for publication once it meets all outstanding technical requirements.

Kind regards,

Farzin Hajebrahimi, Ph.D

Academic Editor

PLOS ONE

Reviewers' comments:

Reviewer's Responses to Questions

**Comments to the Author**

1. If the authors have adequately addressed your comments raised in a previous round of review and you feel that this manuscript is now acceptable for publication, you may indicate that here to bypass the “Comments to the Author” section, enter your conflict of interest statement in the “Confidential to Editor” section, and submit your "Accept" recommendation.

Reviewer #1: All comments have been addressed

Reviewer #2: All comments have been addressed

Reviewer #3: All comments have been addressed

2. Is the manuscript technically sound, and do the data support the conclusions?

Reviewer #1: Yes

Reviewer #2: Yes

Reviewer #3: Yes

3. Has the statistical analysis been performed appropriately and rigorously? 

Reviewer #1: Yes

Reviewer #2: Yes

Reviewer #3: Yes

4. Have the authors made all data underlying the findings in their manuscript fully available?

Reviewer #1: Yes

Reviewer #2: Yes

Reviewer #3: Yes

5. Is the manuscript presented in an intelligible fashion and written in standard English?

Reviewer #1: Yes

Reviewer #2: (No Response)

Reviewer #3: Yes

6. Review Comments to the Author

Reviewer #1: It is unclear why the authors chose not to include a scale or unit for the colorbar in the figures, despite this being an important aspect for interpretation. In their response, they stated, “The scale on the left represents t-values derived from statistical comparisons.” Could the authors clarify if they are referring to the color squares on the left as representing t-values from the statistical comparisons? Including a clear and explicit scale/unit for the colorbar would greatly enhance the clarity and interpretability of the figures.

Reviewer #2: I agree with the changes made to the manuscript, as they effectively address the comments and questions raised in the previous review round. This article sheds light on a relatively unexplored area within prospective memory research, and I believe it will be of great value to the journal's readers and researchers in the field.

Reviewer #3: Thank you for addressing my concerns. The manuscript is technically sound, and data support the conclusions.

7. PLOS authors have the option to publish the peer review history of their article (what does this mean? ). If published, this will include your full peer review and any attached files.

**Do you want your identity to be public for this peer review?** For information about this choice, including consent withdrawal, please see our Privacy Policy .

Reviewer #1: No

Reviewer #2: **Yes: ** Cristina López-Rojas

Reviewer #3: No

---

## [Editor Report · Acceptance letter]

PONE-D-24-30533R1

PLOS ONE

Dear Dr. VICENTIN,

I'm pleased to inform you that your manuscript has been deemed suitable for publication in PLOS ONE. Congratulations! Your manuscript is now being handed over to our production team.

Kind regards,

on behalf of

Dr. Farzin Hajebrahimi

Academic Editor

PLOS ONE